# Efficient and Local Parallel Random Walks

**Michael Kapralov**
EPFL
michael.kapralov@epfl.ch

**Silvio Lattanzi**
Google Research
silviol@google.com

**Navid Nouri**
EPFL
navid.nouri@epfl.ch

**Jakab Tardos**
EPFL
jakab.tardos@epfl.ch

## Abstract

Random walks are a fundamental primitive used in many machine learning algorithms with several applications in clustering and semi-supervised learning. Despite their relevance, the first efficient parallel algorithm to compute random walks has been introduced very recently (Łącki et al.) Unfortunately their method has a fundamental shortcoming: their algorithm is non-local in that it heavily relies on computing random walks out of all nodes in the input graph, even though in many practical applications one is interested in computing random walks only from a small subset of nodes in the graph. In this paper, we present a new algorithm that overcomes this limitation by building random walk efficiently and locally at the same time. We show that our technique is both memory and round efficient, and in particular yields an efficient parallel local clustering algorithm. Finally, we complement our theoretical analysis with experimental results showing that our algorithm is significantly more scalable than previous approaches.

## 1 Introduction

Random walks are key components of many machine learning algorithms with applications in computing graph partitioning [ST04, GKL$^+$21], spectral embeddings [CPS15, CKK$^+$18], or network inference [HMMT18], as well as learning image segmentation [MS00], ranking nodes in a graph [AC07] and many other applications. With the increasing availability and importance of large scale datasets it is important to design efficient algorithms to compute random walks in large networks.

Several algorithms for computing random walks in parallel and streaming models have been proposed in the literature. In the streaming setting, Sarma, Gollapudi and Panigrahy [DSGP11] introduced multi-pass streaming algorithms for simulating random walks, and recently Jin [Jin19] gave algorithms for generating a single random walk from a prespecified vertex in one pass. The first efficient parallel algorithms for this problem have been introduced in the PRAM model [KNP92, HZ96].

In a more recent line of work, Bahmani, Chakrabarti, and Xin [BCX11] designed a parallel algorithm that constructs a single random walk of length $\ell$ from every node in $O(\log \ell)$ rounds in the massively parallel computation model (MPC), with the important caveat that these walks are not independent (an important property in many applications). This was followed by the work of Assadi, Sun and Weinstein [ASW19], which gave an MPC algorithm for generating random walks in an undirected regular graph. Finally, Łącki et al. [ŁMOS20] presented a new algorithm to compute random walks of length $\ell$ from every node in an arbitrary undirected graph. The algorithm of [ŁMOS20] still uses only $O(\log \ell)$ parallel rounds, and walks computed are now independent.

35th Conference on Neural Information Processing Systems (NeurIPS 2021).

From a high level perspective, the main idea behind all the MPC algorithms presented in [BCX11, ASW19, ŁMOS20] is to compute random walks of length $\ell$ by stitching together random walks of length $\ell/2$ in a single parallel round. The walks of length $\ell/2$ are computed by stitching together random walks of length $\ell/4$ and so on. It is possible to prove that such strategy leads to algorithms that run in $O(\log \ell)$ parallel rounds as shown in previous work (this is also optimal under the 1-vs-2 cycle conjecture, as shown in [ŁMOS20]). Note that this technique in order to succeed computes in round $i$ several random walks of length $2^i$ for all the nodes in the network in parallel. This technique is very effective if we are interested in computing random walks from all the nodes in the graph, or, more precisely, when the number of walks computed out of a node is proportional to its stationary distribution. However, this approach leads to significant inefficiencies when we are interested in computing random walks only out of a subset of nodes or for a single node in the graph. This is even more important when we consider that in many applications as in clustering [GLMY11, GS12, WGD13] we are interested in running random walks only from a small subset of seed nodes. This leads to the natural question: *Is it possible to compute efficiently and in parallel random walks only from a subset of nodes in a graph?*

In this paper we answer this question in the affirmative, and we show an application of such a result in local clustering. Before describing our results in detail, we discuss the precise model of parallelism that we use in this work.

**The MPC model.** We design algorithms in the massively parallel computation (MPC) model, which is a theoretical abstraction of real-world system, such as MapReduce [DG08], Hadoop [Whi12], Spark [ZCF+10] and Dryad [IBY+07]. The MPC model [KSV10, GSZ11, BKS13] is the de-facto standard for analyzing algorithms for large-scale parallel computing.

Computation in MPC is divided into synchronous *rounds* over multiple machines. Each machine has memory $S$ and at the beginning data is partitioned arbitrarily across machines. During each round, machines process data locally and then exchange data with the restriction that no machine receives more than $S$ bits of data. The efficiency of an algorithm in this model is measured by the number of rounds it takes for the algorithm to terminate, by the size of the memory of every machine and by the total memory used in the computation. In this paper we focus on designing algorithm in the most restrictive and realistic regime where $S \in O(n^\delta)$ for a small constant $\delta \in (0, 1)$ – these algorithms are called fully scalable.

**Our contributions.** Our first contribution is an efficient algorithm for computing multiple random walks from a single node in a graph efficiently.

**Theorem 1.** *There exists a fully scalable MPC algorithm that, given a graph $G = (V, E)$ with $n$ vertices and $m$ edges, a root vertex $r$, and parameters $B^*$, $\ell$ and $\lambda$, can simulate $B^*$ independent random walks on $G$ from $r$ of length $\ell$ with an arbitrarily low error, in $O(\log \ell \log_\lambda B^*)$ rounds and $\widetilde{O}(m\lambda \ell^4 + B^* \lambda \ell)$ total space.*

Our algorithm also applies to the more general problem of generating independent random walks from a subset of nodes in the graph:

**Theorem 2.** *There exists a fully scalable MPC algorithm that, given a graph $G = (V, E)$ with $n$ vertices and $m$ edges and a collection of non-negative integer budgets $(b_u)_{u \in V}$ for vertices in $G$ such that $\sum_{u \in V} b_u = B^*$, parameters $\ell$ and $\lambda$, can simulate, for every $u \in V$, $b_u$ independent random walks on $G$ of length $\ell$ from $u$ with an arbitrarily low error, in $O(\log \ell \log_\lambda B^*)$ rounds and $\widetilde{O}(m\lambda \ell^4 + B^* \lambda \ell)$ total space. The generated walks are independent across starting vertices $u \in V$.*

The following remark clarifies the effect of parameter $\lambda$ on the number of machines.

**Remark 1.** *The parameter $\lambda$ has nothing to do with the input of the algorithm, but is a trade-off parameter between space and round complexity. It is useful to think of it as $\lambda = n^\epsilon$ for some $\epsilon$ (not necessarily a constant), in which case we get a round complexity of $O(\log \ell \log B^*/(\epsilon \log n)) \leq O(\log \ell/\epsilon)$ and a total memory of $\widetilde{O}(mn^\epsilon \ell^4 + B^* n^\epsilon \ell)$. We can set $\epsilon$ to, for example, $1/\log \log n$, to get nearly optimal total space and $\widetilde{O}(\log \ell)$ round complexity.*

If we compare our results with previous works, our algorithm computes truly independent random walks as [ŁMOS20] does. This is in contrast with the algorithm of [BCX11], which introduces dependent constructs not independent walks. Our algorithm has significantly better total memory than [ŁMOS20], which would result in memory $\Omega(m \cdot B^*)$ for generating $B^*$ walks out of a root

node $r$. This comes at the cost of a slightly higher number of rounds ($\log_\lambda B^*$, a factor that in many applications can be considered constant).

The main idea is to preform multiple cycles of stitching algorithms, changing the initial distribution of the random walks adaptively. More precisely, in an initial cycle we construct only a few random walks, distributed according to the stationary distribution – this is known to be doable from previous work. Then, in each cycle we increase the number of walks that we build for node $r$ by a factor of $\lambda$ and we construct the walks only by activating other nodes in the graph that contribute actively in the construction of the random walks for $r$. In this way we obtain an algorithm that is significantly more work efficient in terms of total memory, compared with previous work.

Our second contribution is to present an application of our algorithm to estimating Personalized PageRank and to local graph clustering. To the best of our knowledge, our algorithm is the first local clustering algorithm that uses a number of parallel rounds that only have a logarithmic dependency on the length of the random walk used by the local clustering algorithm.

**Theorem 3.** *For any $\lambda > 1$, let $\alpha \in (0, 1]$ be a constant and let $C$ be a set satisfying that the conductance of $C$, $\Phi(C)$, is at most $\alpha/10$ and $\mathrm{Vol}(C) \leq \frac{2}{3}\mathrm{Vol}(G)$. Then there is an MPC algorithm for local clustering that uses $O(\log \ell \cdot \log_\lambda B^*) = O(\log \log n \cdot \log_\lambda(\mathrm{Vol}(C)))$ rounds of communication and total memory $\widetilde{O}(m\lambda\ell^4 + B^*\lambda\ell) = \widetilde{O}(m\lambda + \lambda\mathrm{Vol}(C)^2)$, where $B^* := \frac{10^6 \log^3 n}{\eta^2\alpha^2}$, $\ell := \frac{10 \log n}{\alpha}$ and $\eta = \frac{1}{10\mathrm{Vol}(C)}$, and outputs a cluster with conductance $O(\sqrt{\alpha \log(\mathrm{Vol}(C))})$.*

Finally we present an experimental analysis of our results where we show that our algorithm to compute random walk is significantly more efficient than previous work [ŁMOS20], and that our technique scale to very large graphs.

**Additional related works.** Efficient parallel random walks algorithm have also been presented in distributed computing [DSNPT13] and using multicores [SRFM16]. Although the algorithms in [DSNPT13] require a number of rounds linear in the diameter of the graph. The results in [SRFM16] are closer in spirit to our work here but from an algorithmic perspective the challenges in developing algorithms in multicore and MPC settings are quite different. In our setting, most of the difficulty is in the fact that there is no shared memory and coordination between machines. As a result, bounding communication between machines and number of rounds is the main focus of this line of research. From an experimental perspective an advantage of the MPC environment is that it can scale to larger instances, in contrast an advantage of the multicore approach is that it is usually faster in practice for medium size instances. In our case study, where one is interested in computing several local clusters from multiple nodes(for example to detect sybil attack(look at [ACE$^+$14] and following work for applications) the MPC approach is often more suitable. This is due to the fact that the computation of multiple Personalized PageRank vectors at the same time often requires a large amount of space.

Several parallel algorithm have also been presented for estimating PageRank or PersonalizedPageRank [DSGP11, DSMPU15, BCX11], but those algorithms either have higher round complexity or introduce dependencies between the PersonalizedPageRank vectors computed for different nodes. Finally there has been also some work in parallelize local clustering algorithm [CS15], although all previously known methods have complexity linear in the number of steps executed by the random walk/process used in the algorithm(in fact, our method could potentially be used to speed-up this work as well).

**Notation.** We work on undirected unweighted graphs, which we usually denote by $G = (V, E)$, where $V$ and $E$ are the set of vertices and the set of edges, respectively. We also have $n := |V|$ and $m := |E|$, unless otherwise stated. We define matrix $D$ as the diagonal matrix of degrees, i.e., $D_{i,i} = d(v_i)$. Also, we let $A$ be the adjacency matrix, where $A_{i,j} = 1$ if and only if there is an edge joining $v_i$ and $v_j$, and $A_{i,j} = 0$, otherwise.

## 2 MPC random walks

In this section, we present our main result to compute $B^*$ random walks from a single root vertex $r$ up to a length of $\ell$, then we generalize it to multiple sources. As mentioned before, our main idea is to carefully stitch random walks adaptively to activate nodes locally. In the rest of the section, we start by presenting our main theorem and giving a brief overview of our algorithm. We then describe

our stitching algorithm, and analyze the number of random walks we must start from each node so that our algorithms work. Finally, we present the extension of our result to the setting with multiple sources.

**Theorem 1.** *There exists a fully scalable MPC algorithm that, given a graph $G = (V, E)$ with $n$ vertices and $m$ edges, a root vertex $r$, and parameters $B^*$, $\ell$ and $\lambda$, can simulate $B^*$ independent random walks on $G$ from $r$ of length $\ell$ with an arbitrarily low error, in $O(\log \ell \log_\lambda B^*)$ rounds and $\widetilde{O}(m\lambda\ell^4 + B^*\lambda\ell)$ total space.*

### 2.1 Overview of our Algorithm

Here, we explain the frameworks of stitching and budgeting, which are the two key tools making up our algorithm. For simplicity and without loss of generality, we assume that each vertex $v$ has its own machine that stores its neighborhood, its budgets, and its corresponding random walks.[1]

**Remark 2.** *For ease of notation we assume that $\ell = 2^j$ for some integer $j$. One can see that this assumption is without loss of generality, because otherwise one can always round $\ell$ to the smallest power of $2$ greater than $\ell$, and solve the problem using the rounded $\ell$. This affects the complexity bounds by at most a constant factor.*

**Stitching.**

Here, we explain the framework of stitching, which is a key tool for our algorithm. At each point in time the machine corresponding to $v$ stores sets of random walks of certain lengths, each starting in $v$. Each edge of each walk is labeled by a number from $1$ to $\ell$, denoting the position it will hold in the completed walk. Thus, a walk of length $s$ could be labeled $(k + 1, \ldots, k + s)$ for some $k$. Initially each vertex generates a pre-determined number of random edges (or walks of length one) with each label from $1$ to $\ell$. Thus at this point, we would find walks labeled $1, 2, 3, \ldots$ on each machine. After the first round of stitching, these will be paired up into walks of length two, and so we will see walks labeled by $(1, 2), (3, 4), (5, 6), \ldots$ on each machine. After the second round of stitching we will see walks of length $4$, such as $(1, 2, 3, 4)$, and so on. Finally, after the last round of stitching, each machine will contain some number of walks of length $\ell$ (labeled from $1$ to $\ell$), as desired.

At any given time let $W_k(v)$ denote the set of walks stored by $v$ whose first label is $k$ and $B(v, k)$ denotes their cardinality – in the future, we will refer to the function $B$ as the budget. After the initial round of edge generation, $W_k(v)$ consists of $B(v, k)$ individual edges adjacent to $v$, for each $v$ and $k$.

The rounds of communication proceed as follows: in the first round of stitching, for each edge (or length one walk) $e$ in $W_k(v)$, for any *odd* $k$, $v$ sends a request for the continuation of the walk to $z$, where $z$ is the other endpoint of $e$. That is, $v$ sends a request to $z$ for an element of $W_{k+1}(z)$. Each vertex sends out all such requests simultaneously in a single MPC round. Following this each vertex replies to each request by sending a walk from the appropriate set. Crucially, each request must be answered with a *different, independent* walk. If the number of requests for $W_{k+1}(z)$ exceeds $|W_{k+1}(z)| = B(z, k + 1)$, the vertex $z$ declares failure and the algorithm terminates. Otherwise, all such requests are satisfied simultaneously in a single MPC round. Finally, each vertex $v$ increases the length of each of its walks in $W_k(v)$ to two when $k$ is odd, and deletes all remaining walks in $W_k(v)$ when $k$ is even (see Figure 1). For a more formal definition see Algorithm 1.

**Budgeting.**

A crucial aspect of stitching is that the budgets $B(v, k)$ need to be carefully prescribed. If at any point in time a vertex receives more requests than it can serve, the entire algorithm fails. In the case where $B(\cdot, 1)$ follows the stationary distribution, this can be done (see for instance [ŁMOS20]), since the number of requests – at least in expectation – will also follow the stationary distribution. In our setting however, when $B(\cdot, 1)$ follows the indicator distribution of $r$, this is much more difficult. We should assign higher budgets to vertices nearer $r$; however, we have no knowledge of which vertices these are.

In other words, the main challenge in making stitching work with low space and few rounds is to set the vertex budgets $(B(v, k))$ accurately enough for stitching to succeed – this is the main technical contribution of this paper.

---

[1] In reality multiple vertices may share a machine, if they have low degree; or if a vertex has a high degree it may be accommodated by multiple machines in a constant-depth tree structure.

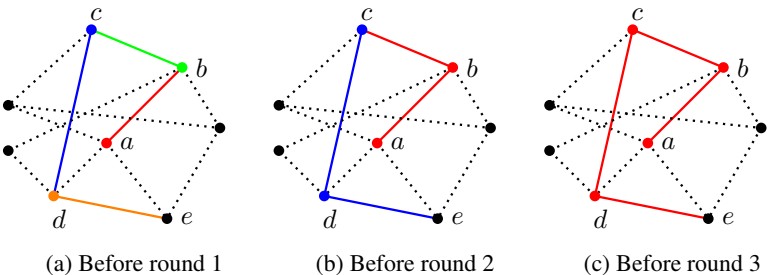

| (a) Before round 1 | (b) Before round 2 | (c) Before round 3 |

Figure 1: Illustration of stitching algorithm for walk $(a, b, c, d, e)$. The red, green, blue and orange walks in each figure correspond to walks in $W_1(a)$, $W_2(b)$, $W_3(c)$ and $W_4(d)$, respectively.

Our technique is to run multiple cycles of stitching sequentially. In the first cycle, we simply start from the stationary distribution. Then, with each cycle, we shift closer and closer to the desired distribution, in which the budget of $r$ is much greater than the budgets of other vertices. We do this by augmenting $B(r, 1)$ by some parameter $\lambda$ each cycle. This forces us to augment other budgets as well: For example, for $u$ in the neighborhood of $r$ we expect to have a significantly increased budget $B(u, 2)$. In order to estimate the demand on $u$ (and all other vertices) we use data from the previous cycle.

Though initially only a few walks simulated by our algorithm start in $r$ (we call these rooted walks), we are still able to derive some information from them. For instance, we can count the number of walks starting in $r$ and reaching $u$ as their second step. If $\kappa$ rooted walks visited $u$ as their second step in the previous cycle, we expect this number to increase to $\lambda \cdot \kappa$ in the following cycle. Hence, we can preemptively increase $B(u, 2)$ to approximately $\lambda \cdot \kappa$.

More precisely, we set the initial budget of each vertex to $\sim B_0 \cdot \deg(v)$ – an appropriately scaled version of the stationary distribution. This guarantees that the first round of stitching succeeds. Afterwards we set each budget $B(v, k)$ individually based on the information gathered from the previous cycle. We first count the number of rooted walks that ended up at $v$ as their $k^{\text{th}}$ step (Line 9). If this number is sufficiently large to be statistically significant (above some carefully chosen threshold $\theta$ in our case, Line 10), then we estimate the budget $B(v, k)$ to be approximately $\lambda \cdot \kappa$ in the following cycle (Line 11). On the other hand, if $\kappa$ is deemed too small, it means that rooted random walks rarely reach $v$ as their $k^{\text{th}}$ step, and the budget $B(v, k)$ remains what it was before.

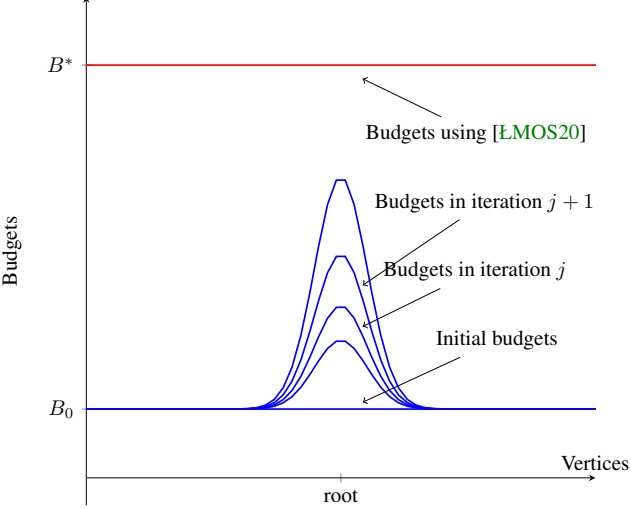

Figure 2: Illustration of the evolution of $B(v, k)$ for all $v \in V$ for a fixed $k$ on a line graph over the iterations of Algorithm 2 and comparing to budgets if one uses [ŁMOS20]. Vertices are sorted by their order on the line and root is the middle vertex on the line.

---

**Algorithm 1** Stitching algorithm

---

1: **procedure** STITCH$(G, B)$
2:   **for** $v \in V$ in parallel **do**
3:     **for** $k \in [\ell]$ **do**
4:       $W_k(v) \leftarrow$ a set of $B(v, k)$ independent uniformly random edges adjacent on $v$
5:   **for** $j = 1 \ldots \log_2 \ell$ **do**                 ▷ See Remark 2
6:     **for** $v \in V$ in parallel **do**
7:       **for** $k \equiv 1 \ (mod\ 2^j)$ **do**
8:         **for** walk $p \in W_k(v)$ **do**
9:           **send** $(v, k)$ **to** $z$, where $z \leftarrow$ end vertex of $p$
10:    **for** $z \in V$ in parallel **do**
11:      **for** each message $(v, k)$ **do**
12:        **if** $W_{k+2^{j-1}}(z) = \emptyset$ **then**
13:          **return Fail**
14:        **send** $(q, k, z)$ **to** $v$, where the walk $q \leftarrow$ a randomly chosen walk in $W_{k+2^{j-1}}(v)$
15:        $W_{k+2^{j-1}}(z) \leftarrow W_{k+2^{j-1}}(z) \backslash \{q\}$
16:    **for** $v \in V$ in parallel **do**
17:      **for** each message $(q, k, z)$ **do**
18:        $p \leftarrow$ any walk of length $2^{j-1}$ in $W_k(v)$ with end vertex $z$
19:        $W_k(v) \leftarrow W_k(v) \backslash \{p\} \cup \{p + q\}$        ▷ $p + q$ is the concatenated walk
20:      **for** $k \equiv 2^{j-1} \ (mod\ 2^j)$ **do**
21:        $W_k(v) \leftarrow \emptyset$
22:    **return** $W_1(v)$ for all $v \in V$

---

**Algorithm 2** Main Algorithm (Budgeting)

---

1: **procedure** MAIN$(G, r, \ell, B^*, \lambda)$
2:   $\theta \leftarrow 10C\ell^2 \log n$, $B_0 \leftarrow 30C\lambda\ell^3 \log n$, $\tau \leftarrow 1 + \sqrt{\frac{20C \log n}{\theta}}$  ▷ **Parameter settings**
3:   $\forall v \in V, B_0(v) \leftarrow B_0 \cdot \deg(v)$
4:   $\forall v \in V, \forall k \in [\ell] : \ B(v, k) \leftarrow B_0 \cdot \deg(v) \cdot \tau^{3k-3}$
5:   **for** $i = 1 \ldots \lfloor \log_\lambda B^* \rfloor$ **do**
6:     $W_1 \leftarrow$ STITCH(G,B)
7:     $W \leftarrow W_1(r)$
8:     **for** $v \in V, \ k \in [\ell]$ **do**
9:       $\kappa \leftarrow |\{w \in W | w_k = v\}|$
10:      **if** $\kappa \geq \theta$ **then**
11:        $B(v, k) \leftarrow (B_0(v) + \lambda^i \cdot \frac{\kappa}{|W|}) \cdot \tau^{3k-3}$
12:      **else**
13:        $B(v, k) \leftarrow B_0(v) \cdot \tau^{3k-3}$
14:   $W_1 \leftarrow$ STITCH$(G, B)$
15:   $W \leftarrow W_1(r)$
16:   **return** $W$

---

**Remark 3.** *In the above pseudocode (Algorithm 2) $\tau$ is a scaling parameter slightly greater then one. We augment all budgets $B(\cdot, k)$ by a factor $\tau^{3k-3}$, to insure that there are always slightly more walks with higher labels, and ensure that stitching succeeds with high probability.*

**Analysis.** We are now ready to present the main properties of our algorithm:

**Lemma 1** (Correctness and complexity). *Algorithm 2 takes $O(\log \ell \cdot \log_\lambda B^*)$ rounds of MPC communication, STITCH terminates without failures with high probability, and the total amount of memory used for walks is $\sum_{v \in V} \sum_{k=1}^{\ell} B(v, k) = O(m\lambda\ell^4 \log n + B^*\lambda\ell)$.*

*Proof.* Let $P^k$ be distribution of random walks of length $k$ starting from $r$. That is, the probability that such a walk ends up in $v \in V$ after $k$ steps is $P^k(v)$.

To bound the round complexity we note that each call of STITCH takes only $O(\log \ell)$ rounds of communication, and it is called $\lfloor \log_\lambda B^* \rfloor + 1$ times; this dominates the round complexity. Further rounds of communication are needed to update the budgets. However this can be done in parallel for each vertex, and thus takes only one round per iteration of the outer for-loop (Line 5).

To prove that the algorithm fails with low probability, we must show the following crucial claim about the budgets $B(V, K)$. Recall that the ideal budget in the $i^{\text{th}}$ iteration would be $B(v, k) \approx B_0(v) + \lambda^i \cdot P^k(v)$. We show that in reality, the budgets do not deviate too much from this.

**Claim 1.** *After iteration $i$ of the outer for-loop (Line 5) in Algorithm 2, with high probability $B$ is set such that*

$$\forall v \in V, \ k \in [\ell] : \ B(v, k) \in \left[ (B_0(v) + \lambda^i \cdot P^k(v)) \cdot \tau^{3k-4}, (B_0(v) + \lambda^i \cdot P^k(v)) \cdot \tau^{3k-2} \right].$$

*Proof.* We first note how $B(r, 1)$ – the budget of walks starting at the root vertex – evolves. For $v = r$ and $k = 1$, $\kappa$ is always equal to $|W|$ – since $r$ is the root vertex – and greater than $\theta$. Therefore, $B(r, 1)$ is set in Line 11 of Algorithm 2 to $(B_0(r) + \lambda^i)$. This is important, because it means that when setting other budgets in iteration $i > 1$, $|W|$ is always $(B_0(r) + \lambda^{i-1})$, the number of walks rooted from $r$ in the previous round. The exception is the first iteration, when $|W|$ is simply $|B_0(r)|$. In both cases we may say that $|W| \geq \lambda^{i-1}$.

There are two options we have to consider: If after the $i^{\text{th}}$ round of STITCH $\kappa$ exceeded $\theta$, in which case our empirical estimator $\kappa/|W|$ for $P^k(v)$ is deemed reliable. We then use this estimater to set the budget for the next round (see Line 11). Alternately, if $\kappa$ did not exceed $\theta$, the imperical estimator is deemed too unreliable; we then simply set $B(v, k)$ proportionally to $B_0(v)$ (see Line 13).

**Case I** ($\kappa < \theta$) then intuitively, $\kappa$ is too small to provide an accurate estimator of $P^k(v)$. In this case we are forced to argue that the (predictable) term $B_0(v)$ dominates the (unknown) term $P^k(v)$. Since $\kappa < \theta$, $\mathbb{E}(\kappa) = P^k(v) \cdot |W| \leq 2\theta$. (The opposite happens with low probability[2] by Chernoff bounds, since $\theta \geq 10C \log n$.) Therefore,

$$B(v, k) = B_0(v) \cdot \tau^{3k-3} \leq (B_0(v) + \lambda^i \cdot P^k(v)) \cdot \tau^{3k-2},$$

and

$$B(v, k) = B_0(v) \cdot \tau^{3k-3} = (B_0(v) + \lambda^i \cdot P^k(v)) \cdot \left( 1 - \frac{\lambda^i \cdot P^k(v)}{B_0(v) + \lambda^i \cdot P^k(v)} \right) \cdot \tau^{3k-3}.$$

So, we need to prove that $1 - \frac{\lambda^i \cdot P^k(v)}{B_0(v) + \lambda^i \cdot P^k(v)} \geq \tau^{-1}$. Now, by the above bound on $\mathbb{E}(\kappa)$ as well as the fact that $|W| \geq \lambda^{i-1}$, we have $2\theta \geq P^k(v) \cdot |W| \geq P^k(v) \cdot \lambda^{i-1}$, which results in $\lambda^i \cdot P^k(v) \leq 2\lambda\theta$. Consequently,

$$\left( 1 - \frac{\lambda^i \cdot P^k(v)}{B_0(v) + \lambda^i \cdot P^k(v)} \right) \geq \left( 1 - \frac{2\lambda\theta}{B_0(v) + 2\lambda\theta} \right) \geq \tau^{-1}.$$

Here we used $B_0(v) \geq B_0 \geq 3\lambda\theta \cdot (\sqrt{\theta/(20C \log n)})$, which holds by definition of $B_0(v)$ and our setting of parameters $B_0$ and $\theta$.

**Case II** ($\kappa \geq \theta$), then intuitively $\kappa$ is robust enough to provide a reliable estimator for $P^k(v)$. More precisely, $\kappa/|W| \in \left[ \mathbb{E}(\kappa/|W|) \cdot \tau^{-1}, \mathbb{E}(\kappa/|W|) \cdot \tau \right]$ with high probability – indeed $\tau$ is defined in terms of $\theta$ deliberately in exactly such a way that this is guaranteed by Chernoff bounds. $\mathbb{E}(\kappa/|W|) = P^k(v)$, therefore

$$\lambda^i \cdot \frac{\kappa}{|W|} \in \left[ \lambda^i \cdot P^k(v) \cdot \tau^{3k-4}, \lambda^i \cdot P^k(v) \cdot \tau^{3k-2} \right],$$

and

$$\begin{aligned} B(v, k) &= (B_0(v) + \lambda^i \cdot \frac{\kappa}{|W|}) \cdot \tau^{3k-3} \\ &\in \left[ (B_0(v) + \lambda^i \cdot P^k(v)) \cdot \tau^{3k-4}, (B_0(v) + \lambda^i \cdot P^k(v)) \cdot \tau^{3k-2} \right]. \end{aligned}$$

$\square$

---

[2]Throughout the proof, we say 'low probability' to mean probability of $n^{-\Omega(C)}$ where $C$ can be set arbitrarily high.

STITCH only reports failure if for some $v \in V$ and $k \in [2, \ell]$, vertex $v$ receives more requests for walks in $W_k(v)$ than $|W_k(v)| = B(v, k)$. Number of such request is upper bounded by the number of edges ending in $v$ generated by neighbors of $v$, say $w$ at level $k - 1$. That is, the number of requests for $W_k(v)$ is in expectation at most

$$\sum_{w \in \Gamma(v)} \frac{1}{d(w)} B(w, k-1) \leq \sum_{w \in \Gamma(v)} \frac{1}{d(w)} (B_0(w) + \lambda^i \cdot P^{k-1}(w)) \cdot \tau^{3k-5}$$

$$= \left( \sum_{w \in \Gamma(v)} \frac{1}{d(w)} B_0(w) + \lambda^i \cdot P^k(v) \right) \cdot \tau^{3k-5}$$

$$= \left( d(v) B_0 + \lambda^i \cdot P^k(v) \right) \cdot \tau^{3k-5}$$

$$= \left( B_0(v) + \lambda^i \cdot P^k(v) \right) \cdot \tau^{3k-5}.$$

Since this is greater than $\theta$, the actual number of requests is at most $(B_0(v) + \lambda^i \cdot P^k(v)) \cdot \tau^{3k-4} \leq B(v, k)$, with high probability by Chernoff. Therefore, STITCH indeed does not fail.

Finally, we prove the memory bound. By setting of parameter $\theta$, $\tau^{3k-2}$ is at most a constant. Also by the setting of parameters we have,

$$B(v, k) \leq (B_0(v) + \lambda^{\lfloor \log_\lambda B^* \rfloor + 1} \cdot P^k(v)) \cdot \tau^{3k-2} = O(B_0(v) + B^* \lambda \cdot P^k(v)),$$

and $\sum_{v \in V} \sum_{k=1}^{\ell} B(v, k) = O(1) \cdot \sum_{v \in V} \sum_{k=1}^{\ell} (B_0(v) + B^* \lambda \cdot P^k(v)) = O(m \lambda \ell^4 \log n + B^* \lambda \ell).$ $\square$

This gives us the proof of Theorem 1. Now, one can easily extend this result to the case when multiple sources for the starting vertex is considered. Theorem 2 is proven in Appendix A.

## 3  From random walks to local clustering

We now present two applications for our algorithm to compute random walks. In particular we show how to use it to compute PageRank vectors and how to use it to compute local clustering. In interest of space, we only state here our main results and we defer all the technical definition and proofs to the Appendix.

**Approximating PageRank using MPC random walks** Interestingly, we can show that we can use our algorithm as a primitive to compute PersonalizedPageRank for a node of for any input vector[3]

**Theorem 4** (Approximating PersonalizedPageRank using MPC random walks). *For any starting single vertex vector $s$ (indicator vector), any $\alpha \in (0, 1)$ and any $\eta$, there is a MPC algorithm that using $O(\log \ell \cdot \log_\lambda B^*)$ rounds of communication and the total amount of memory of $O(m \lambda \ell^4 \log n + B^* \lambda \ell)$, outputs a vector $\widetilde{q}$, such that $\widetilde{q}$ is a $\eta$-additive approximation to $\mathrm{pr}_\alpha(s)$, where $B^* := \frac{10^6 \log^3 n}{\eta^2 \alpha^2}$ and $\ell := \frac{10 \log n}{\alpha}$.*

The proof is deferred to Appendix C.

**Using approximate PersonalizedPageRank vectors to find sparse cuts** Now we can use the previous result on PersonalizedPageRank to find sets with relatively sparse cuts. Roughly speaking, we argue that for any set $C$ of conductance $O(\alpha)$, for many vertices $v \in C$, if we calculate an approximate $\mathrm{pr}_\alpha(v)$ using our algorithms and perform a sweep cut over it, we can find a set of conductance $O(\sqrt{\alpha \log(\mathrm{Vol}(C))})$. This result is stated in Theorem 3. The proof of this result is very similar to the proofs of Section 5 of [ACL06], however since our approximation guarantees are slightly different, we need to modify some parts of the proof for completeness. The full proof is presented in Appendix Our main result of this subsection is stated below.

---

[3]We note that [BCX11] also propose an algorithm to compute PersonalizedPageRank vector but with the limitation that this could be computed only for a single node and not for a vector.

# 4 Empirical Evaluation

In this Section we present empirical evaluations of our algorithms for random walk generation, as well as clustering. As our datasets, we use several real-world graphs form the Stanford Network Analysis Project [LK14, LKF07, LLDM08, KY04, YL12]. The graphs are undirected and come mostly (though not in all cases) from 'Networks with ground truth communities', where the clustering application is most relevant. In order to demonstrate the scalability of our main Algorithm we use graphs of varying sizes, as demonstrated in the table below.

Table 1: Summary of the various graphs used in our empirical evaluations.

| NAME | VERTICES | EDGES | DESCRIPTION |
|---|---|---|---|
| CA-GRQC | 5424 | 14,496 | COLLABORATION NETWORK |
| EMAIL-ENRON | 36,692 | 183,831 | EMAIL COMMUNICATION NETWORK |
| COM-DBLP | 317,080 | 1,049,866 | COLLABORATION NETWORK |
| COM-YOUTUBE | 1,134,890 | 2,987,624 | ONLINE SOCIAL NETWORK |
| COM-LIVEJOURNAL | 3,997,962 | 34,681,189 | ONLINE SOCIAL NETWORK |
| COM-ORKUT | 3,072,441 | 117,185,083 | ONLINE SOCIAL NETWORK |

The experiments were performed on Amazon's Elastic Map-Reduce system using the Apache Hadoop library. The clusters consisted of 30 machines, each of modest memory and computing power (Amazon's `m4.large` instance) so as to best adhere to the MPC setting. Each experiment described in this section was repeated 3 times to minimize the variance in performance inherent in distributed systems like this.

**Practical considerations.** In Section 2 we worked with the guarantee that no walks "fail" in the stitching phase. This assumption can be fulfilled at nearly no expense to the (asymptotic) guarantees in space and round complexity of Theorem 1, and make the proof much cleaner. In practice however, allowing some small fraction of the walks to fail allows for a more relaxed setting of the parameters, and thus better performance.

Each experiment is performed with 15 roots, selected uniformly at random. The main parameters defining the algorithm are as follows: $\ell$ — the length of a target random walk (16 and 32 in various experiments), $C$ — The number of cycles (iterations of the for-loop in Line 5 of Algorithm 2) performed. , $B_0$ — the initial budget-per-degree of each vertex, $\lambda$ — the approximate scaling of the budgets of the root vertices each cycle, $\tau$ — a parameter defining the amount of excess budget used in stitching. This is used somewhat differently here than in Algorithm 2. For more details see Appendix D.

## 4.1 Scalability

In this section we present the results of our experiments locally generating random walks simultaneously from multiple root vertices. We use the graphs COM-DBLP, COM-YOUTUBE, COM-LIVEJOURNAL, and COM-ORKUT, in order to observe how the runtime of Algorithm 2 scales with the size of the input graph. In each of these graphs, 15 root vertices have been randomly chosen. We ran three experiments with various settings of the parameters, of which one is presented below. For additional experiments see Appendix D. $B_0$ is set to be proportional to $n/m$ – that is inverse proportional to the average degree – since the initial budget of each of each vertex is set to $B_0$ times the degree of the vertex.

We report the execution time in the Amazon Elastic Map-Reduce cluster, as well as the number of rooted walks generated. Finally, under 'Walk failure rate', we report the percentage of *rooted* walks that failed in the *last cycle* of stitching. This is the crucial quantity; earlier cycles are used only to calibrate the vertex budgets for the final cycle.[4]

---

[4]For completeness, we report the empirical standard deviation everywhere. Note, however, that due to the high resource requirement of these experiments, each one was only repeated three times.

Table 2: Experiments with $\ell = 16$, $C = 3$, $B_0 = 6n/m$, $\lambda = 32$, $\tau = 1.4$.

| GRAPH | TIME | $B_0$ | ROOTED WALKS GENERATED | WALK FAILURE RATE |
|---|---|---|---|---|
| COM-DBLP | $23 \pm 7$ MINUTES | 1.812 | $96,362 \pm 2597$ | $14.6 \pm 0.6\%$ |
| COM-YOUTUBE | $34 \pm 6$ MINUTES | 2.279 | $53,076 \pm 1185$ | $10.8 \pm 0.5\%$ |
| COM-LIVEJOURNAL | $76 \pm 11$ MINUTES | 0.692 | $184,246 \pm 756$ | $7.9 \pm 0.1\%$ |
| COM-ORKUT | $64 \pm 13$ MINUTES | 0.157 | $200,924 \pm 1472$ | $3.4 \pm 0.0\%$ |

We observe that Algorithm 2 successfully generates a large number of rooted walks – far more than the initial budgets of the root vertices. As predicted, execution time scales highly sublinearly with the size of the input (recall for example, that COM-ORKUT is more than a hundred times larger than COM-DBLP). The failure rate of walks decreases with the size of the graph in this dataset, with that of COM-ORKUT reaching as low as $3.4\%$ on average; this may be due to the the higher average degree of our larger graphs, leading to the random walks spreading out more.

In Appendix D we report the results of two more experiments, including one with longer walks.

### 4.2 Comparison

In this section we compare to the previous work of [ŁMOS20] for generating random walks in the MPC model. This work heavily relies upon generating random walks from all vertices simultaneously, with the number of walks starting from a given vertex $v$ being proportional $d(v)$. In many applications, however, we are interested in computing random walks from a small number of root vertices. The only way to implement this using the methods of [ŁMOS20] is to start with an initial budget large enough to guarantee the desired number of walks from each vertex.

We perform similar experiments to those in the previous section – albeit on much smaller graphs. Each graph has 15 root vertices chosen randomly, from which we wish to sample random walks. In Table 3 we set $B_0$ to 1 and perform $C = 3$ cycles of Algorithm 2 with $\lambda = 10$, effectively augmenting the budget of root vertices by a factor 100 by the last cycle. Correspondingly, we implement the algorithm of [ŁMOS20] – which we call UNIFORM STITCHING – by simply setting the initial budget 100 times higher, and performing only a single cycle of stitching, ie.: $B_0 = 100$, $C = 1$.

Table 3: Experiments with $\ell = 16$, $\lambda = 10$, $\tau = 1.3$. The row labeled Algorithm 2' corresponds to $B_0 = 1$, $C = 3$, while the row labeled 'Uniform Stitching' corresponds to $B_0 = 100$, $C = 1$.

| ALGORITHM | CA-GRQC | EMAIL-ENRON | COM-DBLP |
|---|---|---|---|
| ALGORITHM 2 | $15 \pm 1$ MINUTES | $19 \pm 1$ MINUTES | $18 \pm 1$ MINUTES |
| UNIFORM STITCHING | $7 \pm 0$ MINUTES | $15 \pm 0$ MINUTES | $66 \pm 1$ MINUTES |

We observe that the running time of our Algorithm 2 barely differs across the three graphs, despite the nearly 100-factor difference between the sizes of CA-GRQC and COM-DBLP. At this small size, the execution time is dominated by setting up the 12 Map-Reduce rounds required to execute Algorithm 2 with these parameters. As expected, the baseline far outperforms our algorithm on the smallest graph; as the size of the input graph grows, however, its running time deteriorates quickly. For larger setting of the parameters UNIFORMSTITCHING can no longer complete on the cluster, due to the higher memory requirement, as we can see in Table 6 in Appendix D.

## Acknowledgments and Disclosure of Funding

This project has received funding from the European Research Council (ERC) under the European Union's Horizon 2020 research and innovation programme (grant agreement No 759471).

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
