# A    Proof of Theorem 2

**Theorem 2.** *There exists a fully scalable MPC algorithm that, given a graph $G = (V, E)$ with $n$ vertices and $m$ edges and a collection of non-negative integer budgets $(b_u)_{u \in V}$ for vertices in $G$ such that $\sum_{u \in V} b_u = B^*$, parameters $\ell$ and $\lambda$, can simulate, for every $u \in V$, $b_u$ independent random walks on $G$ of length $\ell$ from $u$ with an arbitrarily low error, in $O(\log \ell \log_\lambda B^*)$ rounds and $\widetilde{O}(m\lambda\ell^4 + B^*\lambda\ell)$ total space. The generated walks are independent across starting vertices $u \in V$.*

*Proof.* First we consider the setting where all budgets $b_u$ are either the same value $b$, or $0$. We call vertices $u$, where $b_u = b$ roots, and the set of roots $R$. We can now run Algorithm 2, with two simple modification: In Line 7 we set $W$ to be all rooted walks, that is $W \leftarrow \cup_{r \in R} W_1(R)$. Correspondingly, in Line 11, we set the budget to $B(v, k) = (B_0(v) + R \cdot \lambda^i \cdot \frac{\kappa}{|W|}) \cdot \tau^{3k-3}$, since there are now $R$ times as many rooted walks.

From here, the proof of correctness proceeds nearly identically. In the case of a single vertex, we defined $P^k(v)$ as the probability that a walk from $r$ reaches $v$ as its $k^{\text{th}}$ step. Here we must define such a quantity for each $r \in R$: $P_r^k(v)$. The analogous claim to the central Claim 1 is that for all $v \in V$ and $k \in [\ell]$:

$$B(v, k) \in \left[ \left( B_0(v) + \lambda^i \cdot \sum_{r \in R} P_r^k(v) \right) \cdot \tau^{3k-4}, \left( B_0(v) + \lambda^i \cdot \sum_{r \in R} P_r^k(v) \right) \cdot \tau^{3k-2} \right].$$

In order to generalize to an arbitrary vector of budgets $(b_u)_{u \in V}$, we simply write $b$ as the summation of vectors $b^{(1)}, \ldots, b^{(\log B^*)}$, where each vector $b_i$ has all of it's non-zero entries within a factor 2 of each other. We then simply augment the coordinates of each $b_i$ where necessary, to get vectors $\widetilde{b}^{(i)}$ which have all non-zero entries equal to each other. At this point we have reverted to the simpler case: we can run our algorithm in parallel for all $\log B*$ budget vector, which incurs the insignificant extra factor of $\log B*$ in memory.

$\square$

# B    Preliminaries of Section 3

For an undirected graph $G = (V, E)$, for each vertex $v \in V$, we denote its degree by $d(v)$ and for any set $S \subset V$, we define $\text{Vol}(S) := \sum_{v \in S} d(v)$ and $\text{Vol}(G) = 2|E|$. We define the stationary distribution over the graph as

$$\forall v \in V : \ \psi(v) := \frac{d(v)}{\text{Vol}(G)}$$

For any vector $p$ over the vertices and any $S \subseteq V$ we define

$$p(S) := \sum_{v \in S} p(v).$$

Moreover for any vector $p$ over the vertices, we define $p_+$ as follows:

$$\forall v \in V : \ p_+(v) = \max(p(v), 0).$$

The *edge boundary* of a set $S \subseteq V$ is defined as

$$\partial(S) = \{\{u, v\} \in E \text{ such that } u \in S, v \notin S\}.$$

The conductance of any set $S \subseteq V$ is defined as

$$\Phi(S) = \frac{|\partial(S)|}{\min\{\text{Vol}(S), 2m - \text{Vol}(S)\}}$$

**PageRank**   In the literature, PageRank was introduced for the first time in [BP98, PBMW99] for search ranking with starting vector of $s = \vec{1}/n$ (the uniform vector). Later, personalized PageRank introduced where the starting vector is not the uniform vector, in order to address personalized search ranking problem and context sensitive-search [Ber07, FR04, Hav03, JW03]. In the rest of this paper we mostly work with personalized PageRanks, where the starting vector is an indicator vector for a vertex in the graph, and we use the general term of PageRank (as opposed to personalized PageRank) to avoid repetition.

**Definition 1** (PageRank). *The PageRank vector $pr_\alpha(s)$ is defined as the unique solution of the linear system $\mathrm{pr}_\alpha(s) = \alpha s + (1 - \alpha)\mathrm{pr}_\alpha(s)W$, where $\alpha \in (0, 1]$ and called the* teleport probability*, $s$ is the* starting vector*, and $W$ is the lazy random walk transition matrix $W := \frac{1}{2}(I + D^{-1}A)$.*

Below, we mention a few facts about PageRank vectors.

**Fact 1.** *For any starting vector $s$, and any constant $\alpha \in (0, 1]$, there is a unique vector $\mathrm{pr}_\alpha(s)$ satisfying $\mathrm{pr}_\alpha(s) = \alpha s + (1 - \alpha)\mathrm{pr}_\alpha(s)W$.*

**Fact 2.** *A PageRank vector is a weighted average of lazy random walk vectors. More specifically, $\mathrm{pr}_\alpha(s) = \alpha s + \alpha \sum_{t=1}^{\infty}(1 - \alpha)^t(sW^t)$.*

Now, we define a notion of approximation that will be used throughout the paper.

**Definition 2.** *($\eta$-additive approximations) We call a vector $q$, an $\eta$-additive approximate PageRank vector for $p := \mathrm{pr}_\alpha(s)$, if for all $v \in V$, we have $q(v) \in [p(v) - \eta, p(v) + \eta]$.*

**Sweeps**   Suppose that we are given a vector $p$ that imposes an ordering over the vertices of graph $G = (V, E)$, as $v_1, v_2, \ldots, v_n$, where the ordering is such that

$$\frac{p(v_1)}{d(v_1)} \geq \ldots \geq \frac{p(v_n)}{d(v_n)}.$$

For any $j \in [n]$ define, $S_j := \{v_1, \ldots, v_j\}$. We define

$$\Phi(p) := \min_{i \in [n]} \Phi(S_i).$$

**Empirical vectors**   Suppose that a distribution over vertices of the graph is given by a vector $q$. Now, imagine that at each step, one samples a vertex according to $q$, independently, and repeats this procedure for $M$ rounds. Let vector $N$ be such that for any vertex $v \in V$, $N(v)$ is equal to the number of times vertex $v$ is sampled. We call vector $\widetilde{q}$ a $(M, q)$-empirical vector, where

$$\forall v \in V : \ \widetilde{q}(v) := \frac{N(v)}{M}$$

**Claim 2** (Additive guarantees for empirical vectors). *Let $q$ be a distribution vector over vertices of graph, where for each coordinate. Then, let vector $\widetilde{q}$ be a $(\frac{100}{\beta^2}\log n, q)$-empirical vector, for some $\beta$. Then $\forall v \in V : \ |q(v) - \widetilde{q}(v)| \leq \beta$ with high probability.*

*Proof.* Using additive Chernoff Bound (Lemma 2 with $N = \frac{100\log n}{\beta^2}$ and $\Delta = \beta$), for any $v \in V$, we have

$$\Pr[|q(v) - \widetilde{q}(v)| > \beta] \leq 2\exp\left(-2\frac{100\log n}{\beta^2}\beta^2\right) \leq n^{-20}.$$

Taking union bound over the vertices of the graph concludes the proof.   $\square$

# C   Omitted claims, proofs and figures

**Lemma 2** (Additive Chernoff Bound). *Let $X_1, X_2, \ldots, X_N \in [0, 1]$ be $N$ iid random variables, let $\bar{X} := (\sum_{i=1}^{N} X_i)/N$, and let $\mu = \mathbb{E}[\bar{X}]$. For any $\Delta > 0$ we have*

$$\Pr[\bar{X} - \mu \geq \Delta] \leq \exp\left(-2N\Delta^2\right)$$

*and*

$$\Pr[\bar{X} - \mu \leq -\Delta] \leq \exp\left(-2N\Delta^2\right).$$

**Proof of Theorem 4:** We prove this theorem in a few steps. First, we prove that a proper truncation of the formula in Fact 2 is a good approximation for PageRank vector:

**Claim 3.** *For $T \geq \frac{10 \log n}{\alpha}$, we have that $q := \alpha s + \alpha \sum_{i=1}^{T} (1-\alpha)^i (sW^i)$ is a $n^{-10}$-additive approximate PageRank vector for $p := \mathrm{pr}_\alpha(s)$.*

*Proof.* Since $s$ is an indicator vector and $W$ is a lazy random walk matrix, for any integer $t > 0$, $sW^t$ is a distribution vector, and consequently every coordinate is bounded by 1. So, for any vertex $v \in V$, we can bound $q(v) - p(v)$ in the following way:

$$|q(v) - p(v)| \leq \alpha \sum_{i=T+1}^{\infty} (1-\alpha)^i \leq (1-\alpha)^{\frac{10 \log n}{\alpha}} \leq \left( e^{-\alpha} \right)^{\frac{10 \log n}{\alpha}} = n^{-10},$$

since $1 - \alpha \leq e^{-\alpha}$ and $T \geq \frac{10 \log n}{\alpha}$. $\qquad\square$

From now on, we set $T := \frac{10 \log n}{\alpha}$. Now, we show that using empirical vectors output by our parallel algorithm for generating random walks incurs small error.

**Claim 4.** *For any $i \in [T]$, let $q_i$ be the distribution vector for the end point of lazy random walks of length $i$, output by the main algorithm with TVD error of $n^{-10}$ (see Theorem 1). Additionally, let vector $\widetilde{q}_i$ be a $\left( \frac{10^6 \log^3 n}{\eta^2 \alpha^2}, q_i \right)$-empirical vector. Now define*

$$\widetilde{q} := \alpha s + \alpha \sum_{i=1}^{T} (1-\alpha)^i \cdot \widetilde{q}_i$$

*for a constant $\alpha \in (0,1)$ and $T = \frac{10 \log n}{\alpha}$. Then, $\widetilde{q}$ is an $\eta$-additive approximation to $p := \mathrm{pr}_\alpha(s)$.*

*Proof.* For the upper bound, for any $v \in V$ we have

$$
\begin{aligned}
\widetilde{q}(v) &= \alpha s + \alpha \sum_{i=1}^{T} (1-\alpha)^i \cdot \widetilde{q}_i(v) \\
&\leq \alpha s + \alpha \sum_{i=1}^{T} (1-\alpha)^i \cdot \left( q_i(v) + \frac{\eta \alpha}{100 \log n} \right) && \text{By Claim 2 with } \beta = \frac{\eta \alpha}{100 \log n} \\
&\leq \alpha s + \alpha \sum_{i=1}^{T} (1-\alpha)^i \cdot q_i(v) + \frac{\eta}{10} && \text{Since } T = \frac{10 \log n}{\alpha} \\
&\leq \alpha s + \alpha \sum_{i=1}^{T} (1-\alpha)^i (sW^i) + n^{-10} + \frac{\eta}{10} && \text{Using the main algorithm with TVD error } n^{-10} \\
&\leq p(v) + 2n^{-10} + \frac{\eta}{10} && \text{By Claim 3} \\
&\leq p(v) + \eta.
\end{aligned}
$$

And similarly for the lower bound, for any $v \in V$ we have

$$\widetilde{q}(v) = \alpha s + \alpha \sum_{i=1}^{T} (1-\alpha)^i \cdot \widetilde{q}_i(v)$$

$$\geq \alpha s + \alpha \sum_{i=1}^{T} (1-\alpha)^i \cdot \left( q_i(v) - \frac{\eta \alpha}{100 \log n} \right) \quad \text{By Claim 2 with } \beta = \frac{\eta \alpha}{100 \log n}$$

$$\geq \alpha s + \alpha \sum_{i=1}^{T} (1-\alpha)^i \cdot q_i(v) - \frac{\eta}{10} \quad \text{Since } T = \frac{10 \log n}{\alpha}$$

$$\geq \alpha s + \alpha \sum_{i=1}^{T} (1-\alpha)^i (sW^i) - n^{-10} + \frac{\eta}{10} \quad \text{Using the main algorithm with TVD error } n^{-10}$$

$$\geq p(v) - 2n^{-10} - \frac{\eta}{10} \quad \text{By Claim 3}$$

$$\geq p(v) - \eta.$$

$\square$

This means that we need to generate $B^* := \frac{10^6 \log^3 n}{\eta^2 \alpha^2}$ random walks of length $\ell := \frac{10 \log n}{\alpha}$. Now, using Lemma 1

1. in $O(\log \ell \cdot \log_\lambda B^*)$ rounds of MPC communication,
2. and with the total amount of memory of $O(m \lambda \ell^4 \log n + B^* \lambda \ell)$

we can generate the required random walks. $\square$

**Theorem 5.** *Let $q$ be an $\eta$-additive approximate PageRank vector for $p := \mathrm{pr}_\alpha(s)$, where $||s_+||_1 \leq 1$. If there exists a subset of vertices $S$ and a constant $\delta$ satisfying*

$$q(S) - \psi(S) > \delta$$

*and $\eta$ is such that*

$$\eta \leq \frac{\delta}{8 \left\lceil \frac{8}{\phi^2} \log(4\sqrt{\mathrm{Vol}(S)}/\delta) \right\rceil \min(\mathrm{Vol}(S), 2m - \mathrm{Vol}(S))},$$

*then*

$$\Phi(q) < \sqrt{\frac{18\alpha \log(4\sqrt{\mathrm{Vol}(S)}/\delta)}{\delta}}.$$

**Proof of Theorem 5:** Let $\phi := \Phi(q)$. By Lemma 3, for any subset of vertices $S$ and any integer $t$, we have

$$q(S) - \psi(S) \leq \alpha t + \sqrt{X} \left( 1 - \frac{\phi^2}{8} \right)^t + 2t \cdot X\eta$$

where $X := \min(\mathrm{Vol}(S), 2m - \mathrm{Vol}(S))$. If we set

$$t = \left\lceil \frac{8}{\phi^2} \log(4\sqrt{\mathrm{Vol}(S)}/\delta) \right\rceil \leq \frac{9}{\phi^2} \log(4\sqrt{\mathrm{Vol}(S)}/\delta),$$

then we get

$$\sqrt{\min(\mathrm{Vol}(S), 2m - \mathrm{Vol}(S))} \left( 1 - \frac{\phi^2}{8} \right)^t \leq \frac{\delta}{4}.$$

This results in

$$q(S) - \psi(S) \leq \alpha \frac{9}{\phi^2} \log(4\sqrt{\mathrm{Vol}(S)}/\delta) + \frac{\delta}{4} + 2tX\eta$$

Now, as we did set $\eta$ such that

$$\eta \le \frac{\delta}{8tX}$$

then since we assumed that $q(S) - \psi(S) \ge \delta$ then

$$\frac{\delta}{2} < \alpha \frac{9}{\phi^2} \log(4\sqrt{\text{Vol}(S)}/\delta),$$

which is equivalent to

$$\phi < \sqrt{\frac{18\alpha \log(4\sqrt{\text{Vol}(S)}/\delta)}{\delta}}.$$

$\square$

**Lemma 3.** *Let $q$ be an $\eta$-additive approximate PageRank vector for $p := \text{pr}_\alpha(s)$, where $||s_+||_1 \le 1$. Let $\phi$ and $\gamma$ be any constants in $[0, 1]$. Either the following bound holds for any set of vertices $S$ and any integer $t$:*

$$q(S) - \psi(S) \le \gamma + \alpha t + \sqrt{X}\left(1 - \frac{\phi^2}{8}\right)^t + 2t \cdot X\eta$$

*where $X := \min\left(\text{Vol}(S), 2m - \text{Vol}(S)\right)$, or else there exists a sweep cut $S_j^q$, for some $j \in [1, |\text{Supp}(q)|]$, with the following properties:*

1. *$\Phi(S_j^q) < \phi$,*

2. *For some integer $t$,*

$$q(S_j^q) - \psi(S_j^q) > \gamma + \alpha t + \sqrt{X'}\left(1 - \frac{\phi^2}{8}\right)^t + 2t \cdot X'\eta,$$

   *where $X' := \min(\text{Vol}(S_j^q), 2m - \text{Vol}(S_j^q))$.*

**Proof of Lemma 3:** For simplicity of notation let $f_t(x) := \gamma + \alpha t + \sqrt{\min\left(x, 2m - x\right)}\left(1 - \frac{\phi^2}{8}\right)^t$. We are going to prove by induction that if there does not exist a sweep cut with both of the properties then equation

$$q[x] - \frac{x}{2m} \le f_t(x) + 2t \cdot \min(x, 2m - x)\eta \tag{1}$$

holds for all $t \ge 0$.

**Base of induction** ($t = 0$): We need to prove that for any $x \in [0, 2m]$, $q[x] - \frac{x}{2m} \le \gamma + \sqrt{\min(x, 2m - x)}$. The claim is true for $x \in [1, 2m - 1]$ since $q[x] \le 1$ for any $x$, so, we only need to prove the claim for $x \in [0, 1] \cup [2m - 1, 2m]$.

**Case I,** $x \in [0, 1]$: For $x \in [0, 1]$, $q[0] = 0$ and $q[1] \le 1$ and $q[x]$ is a linear function for $x \in [0, 1]$. Also $\sqrt{\min(x, 2m - x)} = \sqrt{x}$. Since $\sqrt{x}$ is a concave function then the claim holds for $x \in [0, 1]$.

**Case II,** $x \in [2m - 1, 2m]$: In this case $\sqrt{\min(2m - x, x)} + \frac{x}{2m} = \sqrt{2m - x} + \frac{x}{2m}$, which is a concave function. So we only need to check the end points of this interval. For $x = 2m$, the claim holds since $q[2m] = 1$. Similarly, for $x = 2m - 1$, $q[x] \le 1 \le \sqrt{1} + \frac{2m-1}{2m}$.

So the base of induction holds.

**Inductive step:** Now assume that Equation (1) holds for some integer $t$. We prove that it holds for $t + 1$. We only need to prove that it holds for $x_j = \text{Vol}(S_j^q)$ for each $j \in [1, \text{Supp}(q)]$. Consider any $j \in [1, |\text{Supp}(q)|]$, and let $S := S_j^q$. If property 2 does not hold, then the claim holds. If property 1 does not hold, then we have $\Phi(S) \ge \phi$. Assume that $x_j \le m$ (the other case is similar)

$$q[\mathrm{Vol}(S)] - \frac{x_j}{2m} = q(S) - \frac{x_j}{2m} \qquad \text{Since } S \text{ is a sweep cut of } q$$

$$\le p(S) + |S| \cdot \eta - \frac{x_j}{2m} \qquad \text{By Definition 2}$$

Let $F := \mathrm{in}(S) \cap \mathrm{out}(S)$ and $F' := \mathrm{in}(S) \cup \mathrm{out}(S)$. By Lemma 4,

$$p(S) = \alpha s(S) + (1 - \alpha)\left(\frac{1}{2}p(F) + \frac{1}{2}p(F')\right). \tag{2}$$

Consequently, we have

$$q[x_j] \le p(S) + |S| \cdot \eta$$

$$\le \alpha s(S) + (1 - \alpha)\left(\frac{1}{2}p(F) + \frac{1}{2}p(F')\right) + |S| \cdot \eta \qquad \text{By Equation (2)}$$

$$\le \alpha + \left(\frac{1}{2}p(F) + \frac{1}{2}p(F')\right) + |S| \cdot \eta \qquad \text{By } ||s_+||_1 \le 1 \text{ and } \alpha \in [0, 1]$$

$$\le \alpha + \left(\frac{1}{2}q(F) + \frac{1}{2}q(F') + x_j \eta\right) + |S| \cdot \eta \qquad \text{By Claim 5}$$

$$\le \alpha + \left(\frac{1}{2}q[x_j - |\partial(S)|] + \frac{1}{2}q[x_j + |\partial(S)|] + x_j \eta\right) + |S| \cdot \eta \qquad \text{By definition of } q[\cdot]$$

$$= \alpha + \left(\frac{1}{2}q[x_j - \Phi(S)x_j] + \frac{1}{2}q[x_j + \Phi(S)x_j] + x_j \eta\right) + |S| \cdot \eta \qquad \text{By definition of } \Phi(S)$$

$$\le \alpha + \frac{1}{2}q[x_j - \phi x_j] + \frac{1}{2}q[x_j + \phi x_j] + 2x_j \eta \qquad \text{By concavity of } q$$

$$\le \alpha + \frac{1}{2}f_t[x_j - \phi x_j] + \frac{1}{2}f_t[x_j + \phi x_j] + 2tx_j \eta + \frac{x_j}{2m} + 2x_j \eta \qquad \text{By induction assumption}$$

Therefore

$$q[x_j] - \frac{x_j}{2m}$$

$$\le \alpha + \frac{1}{2}f_t[x_j - \phi x_j] + \frac{1}{2}f_t[x_j + \phi x_j] + 2(t+1)x_j \eta$$

$$= \gamma + \alpha + \alpha t + \frac{1}{2}\left(\sqrt{x_j - \alpha x_j} + \sqrt{x_j + \alpha x_j}\right)\left(1 - \frac{\phi^2}{8}\right)^t + 2(t+1)x_j \eta$$

$$\le \gamma + \alpha(t+1) + \sqrt{x_j}\left(1 - \frac{\phi^2}{8}\right)^{t+1} + 2(t+1)x_j \eta$$

$\square$

**Definition 3.** *For any vertex $u \in V$ and any $v$ in neighborhood of $u$, we define*

$$p(u, v) = \frac{p(u)}{d(u)}.$$

*Also, we replace each edge $(u, v) \in E$ with two directed edges $(u, v)$ and $(v, u)$. Now, for any subset of directed edges $A$, we define*

$$P(A) = \sum_{(u,v) \in A} p(u, v).$$

**Definition 4.** *For any subset of vertices $S$, we define*

$$\mathrm{in}(S) = \{(u, v) \in E | v \in S\}$$

*and*

$$\mathrm{out}(S) = \{(u, v) \in E | u \in S\}$$

**Lemma 4.** *If $p = \mathrm{pr}_\alpha(s)$ is a PageRank vector, then for any subset of vertices $S$,*

$$p(S) = \alpha(S) + (1 - \alpha) \left( \frac{1}{2} p(\mathrm{in}(S) \cap \mathrm{out}(S)) + \frac{1}{2} p(\mathrm{in}(S) \cup \mathrm{out}(S)) \right).$$

**Claim 5.** *Suppose that $q$ is an $\eta$-additive approximate PageRank vector for $p = \mathrm{pr}_\alpha(s)$ (see Definition 2). Then, for any subset of vertices $S$, if we let $F := \mathrm{in}(S) \cap \mathrm{out}(S)$ and $F' := \mathrm{in}(S) \cup \mathrm{out}(S)$,*

$$-2\mathrm{Vol}(S)\eta \leq (q(F) + q(F')) - (p(F) + p(F')) \leq 2\mathrm{Vol}(S)\eta$$

*Proof.* By Definition 4, if we define

$$q(F) = \sum_{(u,v)\in F} \frac{q(u)}{d(u)} \leq \sum_{(u,v)\in F} \frac{p(u) + \eta}{d(u)} \leq \sum_{(u,v)\in F} \frac{p(u)}{d(u)} + \eta|F| = p(F) + \eta|F|.$$

Similarly,

$$p(F) - \eta|F| \leq q(F).$$

If we repeat the same procedure for $F' := \mathrm{in}(S) \cup \mathrm{out}(S)$, we get,

$$p(F') - \eta|F'| \leq q(F') \leq p(F') + \eta|F'|.$$

In order to conclude the proof, we only need to note that

$$|F| + |F'| = 2\mathrm{Vol}(S).$$

$\square$

**Lemma 5** (Theorem 4 of [ACL06]). *For any set $C$ and any constant $\alpha \in (0, 1]$, there is a subset $C_\alpha \subseteq C$ with volume $\mathrm{Vol}(C_\alpha) \geq \mathrm{Vol}(C)/2$ such that for any vertex $v \in C_\alpha$, the PageRank vector $\mathrm{pr}_\alpha(\chi_v)$ satisfies*

$$[\mathrm{pr}_\alpha(\chi_v)](C) \geq 1 - \frac{\Phi(C)}{\alpha}$$

*where $[\mathrm{pr}_\alpha(\chi_v)](C)$ is the amount of probability from PageRank vector over set $C$.*

See [ACL06] for the proof of Lemma 5.

**Lemma 6.** *Let $\alpha \in (0, 1]$ be a constant and let $C$ be a set satisfying*

   1. $\Phi(C) \leq \alpha/10$,

   2. $\mathrm{Vol}(C) \leq \frac{2}{3}\mathrm{Vol}(G)$.

*If $q$ is a $\eta$-additive approximation to $\mathrm{pr}_\alpha(\chi_v)$ where $v \in C_\alpha$ and $\eta \leq 1/(10\mathrm{Vol}(C))$, then a sweep over $q$ produces a cut with conductance $\Phi(q) = O(\sqrt{\alpha \log(\mathrm{Vol}(C))})$.*

*Proof.* Since $q$ is a $\eta$-additive approximation to $\mathrm{pr}_\alpha(\chi_v)$, then using Lemma 5 we have

$$q(C) \geq 1 - \frac{\Phi(C)}{\alpha} - \eta \cdot |C| \geq 1 - \frac{\Phi(C)}{\alpha} - \eta \cdot \mathrm{Vol}(C),$$

since $|C| \leq \mathrm{Vol}(C)$. Combining this with the facts that $\Phi(C)/\alpha \leq \frac{1}{10}$ and $\eta \leq 1/(10\mathrm{Vol}(C))$, we have $q(C) \geq 4/5$, which implies

$$q(C) - \psi(C) \geq \frac{4}{5} - \frac{2}{3} = \frac{2}{15}.$$

Now, Theorem 5 implies that

$$\Phi(q) \leq \sqrt{135\alpha \log(30\sqrt{\mathrm{Vol}(C)})}.$$

$\square$

**Proof of Theorem 3:** The proof is by combining Theorem 4 and Lemma 6. $\square$

# D   Additional Experiments

We present the result of experimentation with longer walks ($\ell = 32$) in Table 4. Similarly to the other cases, the algorithm scales extremely well with the size of the graph. Furthermore, we observe that in the case of the smaller of the graphs (COM-DBLP, COM-YOUTUBE), doubling the walk-length has a relatively small effect on the run-time. This is to be expected, as the number of Map-Reduce rounds performed scales logarithmically in $\ell$ (see Theorem 1). In the larger graphs, this is less evident, as the running time depends more and more on the work-load as opposed to the rounds complexity.

Table 4: Experiments with $\ell = 32$, $C = 3$, $B_0 = 5n/m$, $\lambda = 32$, $\tau = 1.3$.

| GRAPH | TIME | $B_0$ | ROOTED WALKS GENERATED | WALK FAILURE RATE |
|---|---|---|---|---|
| COM-DBLP | $25 \pm 2$ MINUTES | 1.51 | $79,103 \pm 2412$ | $19.4 \pm 1.1\%$ |
| COM-YOUTUBE | $45 \pm 1$ MINUTES | 1.9 | $44,839 \pm 179$ | $7.8 \pm 1\%$ |
| COM-LIVEJOURNAL | $115 \pm 3$ MINUTES | 0.576 | $152,126 \pm 3028$ | $7.9 \pm 0.2\%$ |
| COM-ORKUT | $95 \pm 1$ MINUTES | 0.131 | $163,056 \pm 1612$ | $5 \pm 0.1\%$ |

In Table 5 we see an experiment similar to that of Table 2, but with the parameters $B_0$ and $\tau$ somewhat lowered. We confirm the results on Section 4.1 on the scaling of running time with the size of the graph. The lower parameters allow for faster running time. However, this is at the expense of both the walk failure rate and the number of rooted walks generated. With lower $B_0$ and $\tau$ the vertex budgets ($B(v, K)$ from Section 2) are smaller, and allow for higher relative deviation from the expectation, leading to more walk failure. The running time decrease is not significant, especially in the case of our smaller graphs, and we conclude that the setting of parameters in Table 2 are closer to optimal for most applications.

Table 5: Experiments with $\ell = 16$, $C = 3$, $B_0 = 3n/m$, $\lambda = 32$, $\tau = 1.2$.

| GRAPH | TIME | $B_0$ | ROOTED WALKS GENERATED | WALK FAILURE RATE |
|---|---|---|---|---|
| COM-DBLP | $17 \pm 1$ MINUTES | 0.906 | $23,837 \pm 2210$ | $38.3 \pm 0.7\%$ |
| COM-YOUTUBE | $23 \pm 2$ MINUTES | 1.14 | $15,977 \pm 2298$ | $28.1 \pm 1.7\%$ |
| COM-LIVEJOURNAL | $35 \pm 0$ MINUTES | 0.346 | $57,460 \pm 2104$ | $26.2 \pm 0.5\%$ |
| COM-ORKUT | $33 \pm 1$ MINUTES | 0.079 | $66,715 \pm 1502$ | $21.5 \pm 0.3\%$ |

Finally, in Table 6 we present the results of a comparison experiment, extremely similar to that of Table 3, but with $\lambda$ increased to 20. The discrepancy is even more striking. Increasing the target budget by a factor of 4 produces no measurable difference for Algorithm 2. However, UNIFORM STITCHING is no longer able to complete on the cluster for inputs EMAIL-ENRON and COM-DBLP, due to the high memory requirement (denoted as '—').

Table 6: Experiments with $\ell = 16$, $\lambda = 20$, $\tau = 1.3$. The row labeled 'Algorithm 2' corresponds to $B_0 = 1$, $C = 3$, while the row labeled 'Uniform Stitching' corresponds to $B_0 = 400$, $C = 1$.

| ALGORITHM | CA-GRQC | EMAIL-ENRON | COM-DBLP |
|---|---|---|---|
| ALGORITHM 2 | $15 \pm 1$ MINUTES | $19 \pm 1$ MINUTES | $17 \pm 1$ MINUTES |
| UNIFORM STITCHING | $8 \pm 0$ MINUTES | — | — |

**Implementation details.**   In Algorithm 2, $B(v, k)$ – the budget associated with the $k^{\text{th}}$ step of the random walk – is proportional to $\tau^{3k}$ (see Line 11 and Line 13) which can lead to a factor $\tau^{\theta(\ell)}$ blow-up in space. In theory this is not a significant loss asymptotically, due to the settings of $\tau$ and $\theta$. Nonetheless, in practice, we use a more subtle formula which leads only to a factor $\tau^{\log_2 \ell}$ blow-up, while retaining a similar guarantee on the probability of failure.

Furthermore, in Algorithm 2 (and the intuitive explanation before it) we distinguish between $W_k(v)$ for different $k$. That is walk segments have predetermined positions in the walk, and a request to

stitch to a walk ending in $v$ with its $k^{\text{th}}$ step can only be served by a walk starting in $v$ with its $k + 1^{\text{st}}$ step. This is mostly for ease of understanding and analysis. In the implementation we make no such distinction. Each node simply stores a set of walks of length $2^i$ in the $i^{\text{th}}$ round. The initial budget of each vertex $v$ (at the beginning of the cycle) is set to $\sum_k B(v, k)$, where $B(v, k)$ is still calculated according to the formulas in Line 11 and Line 13 of Algorithm 2 (with the exception of the altered $\tau$-scaling term, as mentioned in the paragraph above).