# OpenReview forum: "Efficient and Local Parallel Random Walks"
_NeurIPS.cc/2021/Conference — NeurIPS 2021 Poster_

### Official Review · Reviewer_vgtt · 2021-06-29

**Rating:** 7
**Confidence:** 4

**Summary:**

The paper introduces an algorithm for computing multiple random walks in parallel. Performance is measured using the MPC model. The authors also show an interesting application to local graph clustering. The proposed algorithm improves upon memory requirements of SOTA methods at the cost a logarithmically many more iterations.

**Ethical Concerns:**

No ethical concerns.

**Limitations And Societal Impact:**

The authors adequately addressed the limitations and potential negative societal impact of their work.

**Main Review:**

I have three major comments that hopefully can be addressed.

0. The authors seem to work on the distributed computer setting. However, for the application of local graph clustering, the target clusters are much smaller than the size of the graph. In fact, for large graphs, the target clusters have constant size and they do not grow with the number of nodes in the graph. This means that local graph clustering algorithms like ACL, take a fraction of a second to terminate and their memory footprint is also constant. For these reasons, I do not think that local graph clustering is the best application for the distributed setting. Even the datasets that you have chosen can fit in RAM of my personal computer. I think the authors should work on demonstrating an application where having multiple machines is necessary.

1. The introduction of the paper is very technical. I could not understand what the authors are talking about until I read the proofs in the appendix. I actually believe that reading the proofs was much easier than trying to figure out what is going on from the main paper. I understand that it is important to mention all your contributions first, but the current text is dry. I had to reach page 4 to figure out what stitching is.

2. For the application of local graph clustering, the authors have missed citing and discussing [1]. In this paper the authors work on shared-memory machines and they use the work-depth model. In [1] the authors follow a different approach for parallelization. In particular, the ACL method is a coordinate solver, thus instead of updating one coordinate at each iteration, the authors in [1] suggest to update as many coordinate as the number of threads, or if the the number of threads is larger than the eligible push nodes, then in [1] they update all eligible nodes. The authors of the submitted paper need to compare their approach with [1] both in theory and in practice.

Minor comments

3. Regarding the conductance guaranteed for your modification of ACL. I think you should simply state the minor points where the proof changes instead of re-writing the whole proof.

4. Typo in line 34. "From a" instead of  "From an".

5. Typo in line 45. "that" instead of "than".

6. Typo in line 244. "we" instead of "wee".

7. Lines 520-521 in the appendix. This sentence seems incomplete.

[1] J. Shun, F. Roosta-Khorasani, K. Fountoulakis, M. W. Mahoney. Parallel Local Graph Clustering. Proceedings of the VLDB Endowment, Volume 9, Issue 12, August 2016 pp 1041–1052.

**Time Spent Reviewing:**

8 hours.

---

> ### Author Response · Authors · 2021-08-09
> **Reply to Reviewer vgtt**
>
> Thank you for your review, we think that we will be able to address your point for the final version of the paper.
>
> Answer to question 0: The application that we have in mind is when multiple clusters have to be computed for multiple seeds nodes at the same time. This is a quite common problem in real world applications(see for instance [GS12]) and in this case distributed computing can be used to compute multiple clusters at the same time. Note that also in the experimental section we compute multiple clusters at the same time.
>
> Answer to question 1: Thanks for the suggestion, we will move a high level description of the algorithm in the introduction
>
> Answer to question 2: Thanks for raising this point, we will add a reference to [1]. We will add a discussion on the multi-threaded approach although we think that the two approaches are working on different frameworks and addressing different aspects of the problem.
>
> We will also address all the minor comments.

---

> > ### Comment · Reviewer_vgtt · 2021-08-09
> > **Reply**
> >
> > Ok thanks. Some advice that will you help you increase the score of your paper.
> >
> > It is not enough to simply say that "we think that the two approaches are working on different frameworks and addressing different aspects of the problem". In fact, this answer does not offer an proper justification. This way you are asking me to increase your score based on the premise that you will deliver something that you are not even certain about.
> >
> > The way to reply to this is to provide a fully justified argument about the claim that you just made. Since, the reviewing system changed, you might have a chance to defend your paper better.
> >
> > Same holds for the high level description, you have to at least explain how you are going to change the paper.

---

> > > ### Author Response · Authors · 2021-08-10
> > > **Replay**
> > >
> > > Thanks for your suggestion and request for clarifications.
> > >
> > > For point 1, we plan to modify the introduction and to add the description of the main algorithm and intuitions about the proofs there. We will mostly focus on the stitching and budgeting argument. We think that this will also clarify the difference from previous work.
> > >
> > > For point 3, we will clarify the difference between the two systems.
> > > In particular, from an algorithmic perspective the challenges in developing algorithms in the two frameworks are quite different. In a distributed setting, most of the difficulty is in the fact that there is no shared memory and coordination between threads so bounding communication between machines and number of rounds is the main focus of this line of research.
> > > From an infrastructure perspective, the decision of using a distributed environment or a multi-threaded one depends on many external factors. One advantage of the distributed environment is that it can scale to larger instances, in contrast an advantage of the multi-threaded one is that it is usually faster in practice. For our application, when one is interested in computing several local clusters from multiple nodes(for example to detect sybil attack(look at [1] and following work)) we think that the distributed system is often more suitable. This is due to the fact that to compute multiple Personalized PageRank vectors at the same time it often requires a lot of space.
> > > From an experimental point of view, it would be harder to compare the two methods because they require different infrastructure to be run on. Multi-threaded algorithms usually require a single stronger machine and distributed ones require multiple weaker machines. So it is a bit hard to compare the methods in a fair way.
> > > We will add these clarifications in the final version of the paper.
> > >
> > > Please let us know if you have any additional questions and sorry if the previous answers were not very satisfying.
> > >
> > > Thanks again
> > >
> > > Citation:
> > > [1] Lorenzo Alvisi, Allen Clement, Alessandro Epasto, Silvio Lattanzi, Alessandro Panconesi: SoK: The Evolution of Sybil Defense via Social Networks. IEEE Symposium on Security and Privacy 2013: 382-396

---

### Official Review · Reviewer_a4aA · 2021-07-17

**Rating:** 5
**Confidence:** 5

**Summary:**

This paper studies the problem of parallelizing single-source and subset random walk computations. The primary technical contribution is an algorithm with the number of parallel rounds that depends on the length of the random walk in a logarithmic way. In addition, two applications are discussed in the article: personalized PageRank estimation and local clustering.

**Limitations And Societal Impact:**

There is no evidence that this work will have a detrimental effect on society.

**Main Review:**

Weak points:

W1. The presentation of the paper needs significant improvement. First, there are numerous confusing statements, spelling problems, and typos, making it impossible for me to determine the paper's contributions. More comments can be found under "Minor Comments."  Second, the work lacks the intuition needed to describe the algorithms and proofs, making it nearly impossible for non-experts to comprehend. For example, what is the purpose of k in the Stitching algorithm? What's the difference between odd and even k? More in-depth explanations are anticipated. Finally, some important sections and proofs have been left out. The concluding section, for example, is missing. The preliminaries have been moved to the appendix. The formal version of Theorem 1 is missing. Furthermore, the proof of Theorem 1 has been omitted.

W2. Some proofs aren't as rigorous as others:

(1) Does Remark 1 require the existence of B*<n in order to derive the round complexity: O(log{\ell}log{B*}/(eps*logn))<O(log{\ell}/eps) ?

(2) The error analysis appears to be missing in the proof of Theorem 6. Why is it that B* random walks can guarantee a low TVD error?

(3) Based on the stationary distribution of random walks, the Budgeting algorithm initializes B(v,k) as B_0\*deg(v). The stationary distribution pr_alpha=alpha\*s*(I-(1-alpha)*W)^{-1} is difficult to obtain for the Personalized PageRank. How should we initialize B(v,k) in the context of Personalized PageRank?

W3. There are a few key experiments that are lacking:

(1) The paper only compares the proposed technique to one baseline (LMOS20) in Section 4.2. On both the GRQC and ENRON datasets, however, LMOS20 beats the proposed technique. Furthermore, the results on LiveJournal and Orkut are omitted.  Thus, the empirical effect of the suggested method is difficult to justify.

(2) Experiments for finding local sparse cuts and approximating Personalized PageRank should be provided.


Minor comments:

There are various mistakes and confusing descriptions in the present version. A list of minor comments is shown below.

M1: In Remark 1, lambda should be replaced with n^eps in the total memory complexity.


M2: In Theorem 3, the definition of \Phi(C) is omitted.


M3: The definition of eps is omitted in Theorem 4. Furthermore, eps appears to be irrelevant for obtaining Theorem 4's complexity results.

M4: The assumption "each vertex v has its own machine that stores its neighborhood" is stated twice in lines 136 and 142. One of them can be removed.

M5: can simulate -> we can simulate in Theorems 2 and 4.

M6: Line 85, Page 2: dependen -> dependent.

M7: Line 105, Page 3: then -> than

M8: Line 123, Page 3: As mentioned before our main idea -> As mentioned before, our main idea…

M9: Lines 135, Page 3, and 142, Page 4: assume that -> we assume that.

M10: Line 148, Page 4: After the initial round of edge generation W_k(v) consists of … -> After the initial round of edge generation, W_k(v) consists of ….

M11: Line 159 and line 178, Page 4: There are two headings both named as "Budgeting".

M12: Line 180, Page 4: and to do we initialize -> and we initialize.

M13: Line 182, Page 4: The key observation is that to do it we need to

M14: Line 189, Page 4: thee -> the.

M15: Line 270, Page 8: machines each of modest memory … -> machines. Each of them has modest memory …

M16: Line 473, Page 13: of the supplementary material: the formal definition of TVD error is omitted.

**Time Spent Reviewing:**

8h

---

> ### Author Response · Authors · 2021-08-09
> **Reply to Reviewer a4aA**
>
> We thank the reviewer for the effort and we are sorry for any issue with the current presentation. Using the comment we think that we can greatly improve the presentation quality for the camera ready version.
>
> We will also add additional explanations for the algorithms. In the Stitching algorithm k is the length of the constructed walk and we stitch walks of length equal to a power of 2 + 1(this is the condition in line 7).
>
> We remark that no proofs are left out, some are moved to the Appendix but all the statements are proved in the paper. For example, the formal version of Theorem 1 is Theorem 4 and a formal proof for it is presented.
>
> Answers to additional questions in w2:
>
>      1.) B*<n is not required.
>      2.) Note that the goal of Theorem 6 is to simulate B* random walks. The TVD bound follows from the bound on the failure probability of         ‘stitching’, since ‘stitching’ is guaranteed to produce  a walk from the correct distribution when it succeeds. This is identical to the proof of           Theorem 4.
>     3.) We do not simulate the page rank vector by starting from the stationary distribution of page rank. Instead we use our result on random walk simulation as a black box: we simply simulate walks of a sufficient length, then truncate them according to the appropriate distribution. All of this is described in full in Claim 3, in Appendix C.
>
> Answers to questions w3:
>
> Note that we do not report the result for LMOS20 for LiveJournal and Orkut because the algorithm was unable to run on such large graphs. Similarly, in Appendix D, we report a different setting of the parameters where the algorithm of LMOS20 was even unable to run on Enron and DBLP. The baseline beats our Algorithm on the smallest datasets, but once we scale to larger graphs we see a huge discrepancy in our favor. As stated in Section 4, this is because at small sizes the runtime is dominated by the setting up of the map-reduce computations.
>
> Finally we will address all the typos described in the review in the camera ready version.

---

> > ### Comment · Reviewer_a4aA · 2021-08-31
> > **Raising my score to 5**
> >
> > Thanks for the reply! The rebuttal has partially addressed my concerns, so I am raising my score to 5.

---

### Official Review · Reviewer_oy5A · 2021-07-20

**Rating:** 7
**Confidence:** 4

**Summary:**

This paper studies the problem of implementing random walks in the Massively Parallel Computation (MPC) model using logarithmic number of rounds.  Very recently, [LMOS20] develop a MPC algorithm that using O(log \ell) rounds, computes (independent) random walks of length up to \ell from every node in an arbitrary undirected graph. Their main idea is based on stitching: In order to compute walks of length r, we can stick together two walks of length r/2 and we can do this stitching procedure recursively using O(log \ell) rounds. One issue with this algorithm is that if we want to do a few walks starting from a few nodes, this algorithm still implements all walks for all nodes. In particular, for say B walks of length \ell, the algorithm of [LMOS20] uses O(mB) total memory (for all machines).

The current paper claims to solve this issue by proposing a MPC algorithm that uses a total memory of O(mn^{eps}\ell^4+Bn^{eps}\ell) and computes B walks of a given vertex r. The number of communication rounds of this new algorithm is slightly more and is O(log(\ell)/eps).

**Main Review:**

The paper is written poorly. I spent 7-8 hours to read the paper, but I don’t understand the main algorithm. In particular, the budgeting section which is the novelty of this paper is not clear to me. The overview of Algorithm 2 is very unclear and the pseudocode of Algorithm 2 uses lots of notations that have not been properly defined and is very complicated to understand it. Overall, the claims in this paper are nice, but I cannot verify them using this writeup.

Comments:

1- Line 36, there is a typo: “of of ...“
2- The parameters and notations used in this paper are not clearly defined. For example, in Line 66, Theorem 1, for the first time m is used, but it doesn’t say what m is. In the standard graph theory notation n and m are the number of vertices and edges in. A graph, I am guessing this should be the case here, but it can be written explicitly.
3- I don’t understand why we have an informal version of theorems in this paper. Theorem 1 is the same as Theorem 4. Why is one of them informal and the other formal. Why Theorem 2 is informal?
4- In Remark 1 (Line 78), in the total memory once \Lambda is replaced with n^{eps} and next to it is not replaced.
5- Lines 159 and 178, there are two paragraphs for Budgeting.
6- Line 146, the definitions W_k(v) and  B(v,k) are not clear. B(v,k) is defined in this line, but then in line 159, for the first time is written that B(v,k) is a budget. Why didn’t you define this in Line 146?
7- Section 2.1 gives an overview of the algorithm, but this overview is not precise. In Line 150 they explain the stitching procedure, but they only explain the first round.
8- What is \tau in Algorithm 2.
9- The stitching algorithm (Algorithm 2) is very complicated and long. You can split this procedure to subroutines where the machines requests for walks (v,k) and the the surboutines where the other machines respond.
10- Algorithm 2 is very hard to read. What is line 7 in this algorithm? Using \kappa in Line 9 and above it k are very confusing.

**Time Spent Reviewing:**

8 hours

---

> ### Author Response · Authors · 2021-08-09
> **Reply to Reviewer oy5A**
>
> We thank the reviewer for the effort and we are sorry for any issue with the current presentation. Using the comment we think that we can greatly improve the presentation quality for the camera ready version. In particular, we will clarify Algorithm 2 and we will address all the typos and suggestions from the reviewer.

---

> > ### Comment · Reviewer_oy5A · 2021-08-31
> > **Final comment**
> >
> > Great to hear that the authors will polish the paper. I like the paper and I will increase my score to 7.

---

### Official Review · Reviewer_oXY3 · 2021-07-21

**Rating:** 8
**Confidence:** 4

**Summary:**

This submission gives an algorithm in the MPC model (Massively Parallel Computation) for computing independent random walks from a single vertex or a subset of vertices. A previous paper, [LMOS], on the topic gave an algorithm that allowed for generating random walks from all vertices simultaneously (more specifically, the starting points of generated walks were from the stationary distribution). Suppose that you want to generate many random walks from a single vertex. [LMOS] would require generating lots of random walks from all vertices and because of that would require a lot of total space available in the system, and therefore, it would require a lot of machines. The algorithm introduced in this submission can be significantly more space efficient in this type of setting.


**Ethical Concerns:**

None.

**Limitations And Societal Impact:**

There is no concern about negative social impact.

**Main Review:**


The solution follows the usual idea of stitching shorter random walks in order to create longer random walks. In consecutive rounds, random walks of length 2^k are turned into random walks of length 2^{k+1}. An obstacle that any such method has to overcome is that some of the length 2^k random walks may not have extensions. [LMOS] overcame this obstacle by using the stationary distribution, so the rough number of extensions that are needed is known in advance at the cost of large total space requirement. To avoid this, the paper runs several attempts at creating random walks and adjust based on them the distribution from which short random walks are generated. Overall, this solution is not very difficult in hindsight but it is neat and its simplicity allows for implementing the algorithm.

The submission also explores applications of random walks to computing Personalized PageRank and clustering. Furthermore, it shows experiments in which it positively compares the performance to [LMOS].

Apart from typos and small grammatical errors, the paper is well written and presents the main ideas very clearly.

In my opinion, the paper addresses a well-motivated question with real world applications in a practical model of computation. It would be great to see this paper in NeurIPS.

----
I suggest that the authors spend some time proofreading the paper and fixing spelling and small grammatical errors.

l.25: has -> have

l.34: an high -> a high

l.41: "a random walk" or "random walkS"

l.50: in details -> in detail

page 4: Two paragraphs are titled "Budgeting". Was this intended?

Page 3 and 4 have the same footnote.

wee -> we

----------------

Edit: I acknowledge reading the rebuttal from the authors.


**Time Spent Reviewing:**

8

---

> ### Author Response · Authors · 2021-08-09
> **Reply to Reviewer oXY3**
>
> Thank you for the suggestions, we will proofread the paper carefully and correct the typos.

---

### Decision · Program_Chairs · 2021-09-27

**Decision:**

Accept (Poster)

**Comment:**

The reviewers uniformly liked the algorithmic contribution of the paper. However, there were serious concerns about the writing quality of the paper. The reviewers strongly encourage the authors to polish the paper.
Overall, the quality of the contributions were deemed to be of significant interest to warrant an accept.